# Protein/Fiber Index Modulates Uremic Toxin Concentrations in Hemodialysis Patients

**DOI:** 10.3390/toxins14090589

**Published:** 2022-08-27

**Authors:** Manon Ebersolt, Tacy Santana Machado, Cecilia Mallmann, Nathalie Mc-Kay, Laetitia Dou, Dammar Bouchouareb, Philippe Brunet, Stéphane Burtey, Marion Sallée

**Affiliations:** 1Centre de Néphrologie et Transplantation Rénale, AP-HM, Hôpital de la Conception, 147 Bd Baille, 13005 Marseille, France; 2Aix Marseille Univ, INSERM, INRAE, C2VN, 13005 Marseille, France; 3Centre D’investigation Clinique, Hôpital de la Conception, 13005 Marseille, France

**Keywords:** chronic kidney disease, protein fiber index, uremic toxins, hemodialysis

## Abstract

Background: Indoxyl sulfate (IS) and p-cresyl sulfate (PCS), two uremic toxins (UTs), are associated with increased mortality in patients with chronic kidney disease (CKD). These toxins are produced by the microbiota from the diet and excreted by the kidney. The purpose of this study was to analyze the effect of diet on IS and PCS concentration in hemodialysis (HD) patients. Methods: We performed a prospective monocentric study using a seven-day diet record and determination of serum IS and PCS levels in HD patients. We tested the association between toxin concentrations and nutritional data. Results: A total of 58/75 patients (77%) completed the diet record. Mean caloric intake was 22 ± 9.2 kcal/kg/day. The protein/fiber index was 4.9 ± 1.8. No correlation between IS or PCS concentration and protein/fiber index was highlighted. In the 18 anuric patients (31%) in whom residual renal function could not affect toxin concentrations, IS and PCS concentrations were negatively correlated with fiber intake and positively correlated with the protein/fiber index. In a multivariate analysis, IS serum concentration was positively associated with the protein/fiber index (*p* = 0.03). Conclusions: A low protein/fiber index is associated with low concentrations of uremic toxins in anuric HD patients. Diets with an increased fiber intake must be tested to determine whether they reduce PCS and IS serum concentrations.

## 1. Introduction

Chronic kidney disease (CKD) is a frequent medical condition, with a global prevalence estimated to be between 11 and 13% [1]. CKD and its complications worsen over time, affecting both patients’ quality of life and their survival rate [2]. In its earliest stage, CKD is a risk factor for cardiovascular disease regardless of other traditional cardiovascular risk factors [1,3,4]. Cardiovascular mortality in dialysis patients is five times higher than in the general population, suggesting that uremic toxins (UTs) contribute to this increased risk [5,6]. UTs are excreted in the urine by the kidneys and accumulate in the blood and tissues in kidney failure. Indoxyl sulfate (IS) and p-cresol sulfate (PCS) are protein-bound UTs derived from the action of proteolytic fermentation by colon microbes on amino acids that escape digestion in the small intestine. After gut absorption of indole and paracresyl derived from tryptophan and phenylalanine/tyrosine, respectively [7,8], these derivatives are converted in the liver to IS and PCS. This process is called the diet–gut–liver–kidney axis [9]. High serum concentrations of PCS and IS are associated with vascular injury [10,11], cardiovascular disease, and increased mortality in patients with CKD [5,6,12].

Controlling the concentration of UTs has become a major issue in the management of patients with CKD and particularly dialysis patients. Because IS and PCS are bound to plasma proteins, they are poorly cleared by hemodialysis (HD) [13]. The colonic microbiota has a determining role in the serum concentration of IS and PCS [14]. IS and PCS are not detectable in patients who have undergone colectomy [14]. Recent studies indicate that the intestinal microbiota is modified in dialysis patients [15,16]. This can be explained by uremia and dietary changes that cause dysbiosis and can modulate UT concentrations [17]. In humans, the impact of diet on the microbiota seems to be more important than the impact of CKD itself [18]. Diet can influence UT concentrations. Vegetarians have a decreased production of IS and PCS [19,20]. Therefore, diet seems a useful tool to reduce uremic toxin concentrations.

We conducted this study to evaluate the influence of diet on IS and PCS concentrations in HD patients.

## 2. Results

### 2.1. Patient Characteristics

Seventy-five patients were included in the initial protocol. Of the 75 participants selected for the study, 58 (77%) answered the first questionnaire as illustrated in a flow diagram Figure 1. 

Of the 17 patients who did not complete the questionnaire, 2 partially completed and 15 did not complete the diet record, stating that it was too demanding. Of the 17 patients who did not complete the questionnaire, 5/17 (30%) were anuric. Demographics, biochemical characteristics, and dietary intake of the 58 patients are outlined in Table 1 and Table 2. A total of 64% (37/58) were men with an average age of 66 ± 12 years. Eighteen patients (31%) were anuric. The average energy intake was 22 ± 9.2 kcal/kg/d: 42% carbohydrates, 18% proteins, and 37% lipids. Protein intake (0.95 ± 0.3 g/kg/day) was mainly of animal origin (70% vs. 30%) with an average of 0.64 ± 0.2 g/kg/day (44.6 ± 13.2 g/day) of animal protein vs. 0.28 ± 0.1 g/kg/day (18.5 ± 7.4 g/day) of plant protein. Daily fiber intake was, on average, 0.22 ± 0.11 g/kg/day (14.6 ± 5.2 g/day). The protein/fiber index was 4.9 ± 1.8. Lipid intake consisted of 23.8 ± 9.5 g/day saturated fatty acid and 8.2 ± 4.7 g/day unsaturated fatty acid. 

### 2.2. Uremic Toxins and Diet

We found a significant correlation between IS and residual diuresis, age, dialysis vintage, and serum protein. However, no relation between fiber intake or protein/fiber index and UT concentrations (Appendix A) was highlighted. In this population, there is a weak negative correlation between protein intake in g/kg/day and IS concentration (r = −0.28, *p* = 0.03). In the second step, we focused on anuric patients. The residual renal function could increase the excretion of UTs in an unpredictable way and modify UT serum concentration. The clinical and biological characteristics of anuric patients are compared with non-anuric patients in Table 1 and Table 2. No major differences between anuric and non-anuric patients were observed, except with BMI and dialysis vintage. IS and creatinine concentrations were higher in anuric patients. PCS and serum potassium were no different. Patients in both groups reported low consumption of energy, protein, and fiber. Diet consumption of energy, protein, and fiber was lower in anuric patients and the protein/fiber index was not different. Potassium intake was also lower in the anuric group. In anuric patients, total dietary fiber in g/kg/day was significantly negatively correlated with IS and PCS concentrations (respectively r = −0.56, r = −0.47, *p* < 0.01). No significant association was observed between total protein intake and either uremic toxin. The protein/fiber index was significantly positively correlated with both IS and PCS with, respectively, r = 0.54 and 0.55, *p* < 0.05 (Table 3 and Figure 2). 

We observe a significant association between protein animal/fiber index and IS and a statistical trend with PCS (respectively, r = 0.47 *p* = 0.046 and r = 0.46 *p* = 0.05) but not with the plant protein/fiber index. In univariate analysis, IS and PCS were not associated with known predictors of toxin concentration including age, weight, and dialysis vintage as in the whole population (*p* > 0.05).

In multivariate analysis (Table 4) including known predictor factors (age, dialysis vintage, and weight), we found a positive correlation between IS concentration and the protein/fiber index (*p* = 0.03). PCS concentration was not correlated.

## 3. Discussion

We demonstrate a positive correlation between IS and PCS concentration and the protein/fiber index in anuric dialysis patients. Rossi et al. reported in 40 CKD patients that the protein/fiber index was associated with IS and PCS concentration independently of renal function, diabetes, and gender [21]. In our 18 anuric patients, we demonstrated a negative correlation between IS and PCS concentration and total fiber intake in g/kg/day, and a positive correlation between PCS, IS, and the protein/fiber index. These results suggest that a high-fiber diet with a low protein/fiber ratio could lower UT concentrations in HD patients. This correlation could not be confirmed in the whole study population. The choice to restrict our population to anuric dialysis patients is explained by the fact that in the absence of residual renal function, toxin levels can only be explained by variations in production variations. A negative correlation between residual renal function and IS and PCS concentrations in HD patients was reported in the literature [22]. In the dialysis population, serum creatinine is not associated with residual renal function but rather with protein nutritional status [23]. Our paper reinforces the impact of diet on UT levels and represents a shift in toxin reduction therapy. In recent years, studies aimed at reducing UTs levels have focused on dialysis techniques [24,25] and/or pro/pro/symbiotic strategies [26,27,28,29]. With our study, as well as the results of Rossi et al. [21], the role of dietary intake and the quality of dietary intake on the UT level is reinforced. Reducing tryptophan and/or phenylalanine/tyrosine intakes, which seems the most obvious in view of UT production, is not necessarily the best solution. In fact, reducing tryptophan intake is tantamount to reducing protein intake, which is all the more complicated as patients often already have protein intakes below the recommended level (1.2 g/kg/d) [30]. In addition, Wyant et al. recently identified another tryptophan-derived UT, kynurenic acid, which plays a protective role against ischemia [31]. Reduced tryptophan intake could lead to a decrease in kynurenic acid and worsen ischemic disease, which is a major complication of CKD patients. Reducing deleterious UTs such as IS or PCS while maintaining protective kynurenic acid by increasing fiber intake is certainly a better approach. These results suggest that a high-fiber diet with a low protein/fiber ratio could lower UT concentrations in HD patients.

The effect of nutrients on uremic toxin concentrations could be explained by various mechanisms including the modification of microbiota, the modification of digestive transit time, and their impact on intestinal integrity. Increased fiber intake induces an increase in microbiota mass and shifts the relative abundance of Bacteroidetes and Firmicutes leading to saccharolytic fermentation [32]. Conversely, increased protein intake induces proteolytic fermentation, nitrogen fermentation, and phenol and indole formation by intestinal bacteria [7]. The protein/fiber index could reflect the balance between saccharolytic and proteolytic fermentation. Fiber consumption accelerates digestive transit time. A rise in transit time increases urine phenol excretion, reflecting increased colonic production and absorption of PCS [7,33]. Finally, fiber intake may play a role in restoring the integrity of dysfunctional intestinal barriers with disruption of epithelial tight junction in HD patients [34]. 

Studies of diet impact on UT concentration remain sparse. In healthy volunteers [19] and in HD patients [20], a vegetarian diet is associated with lower concentrations of IS and PCS. The positive impact of a vegetarian diet is due to its high fiber content and resistant starch. Resistant starch decreases the fecal concentration of phenol (PCS precursor) in healthy volunteers which reflects a decline in its production [35]. Trials on fiber-enriched diets in HD patients rely on prebiotics. The use of soluble fiber in HD patients showed a 20 to 30% decrease in PCS concentration without impact on IS concentration [26]. Resistant starch can reduce IS concentration [28,29]. These discordances between the effects of different types of fiber on toxin concentrations confirm that the two toxins are associated with different microbiota [18]. In HD patients, PCS and IS concentrations are associated with increased mortality [36,37]. Increased fiber intake by alimentation rich in fruits and vegetables is associated with lower mortality in the general population [38] and in CKD patients [39,40,41,42]. In dialysis patients, Kalantar et al. reported an increase in mortality if fiber intake was lower than 15 g/day [43]. We believe that a decreased protein/fiber index would be beneficial in HD patients or CKD patients. One of the limitations of a high-fiber diet in HD patients is hyperkalemia. Low fiber intake is associated with low potassium intake perhaps consistent with current dietary counseling avoiding fruits and vegetables to control potassium concentrations [44]. Serum potassium in HD patients depends on several factors other than potassium consumption. There is a weak correlation between potassium intake and serum potassium in chronic dialysis patients [45,46]. In cohort studies, higher consumption of fruits and vegetables is associated with lower mortality in HD patients [47] and in peritoneal dialysis patients [48]. However, fruits and vegetables are not the only sources of potassium.

Our study confirms that dialysis patients have inadequate dietary energy and protein intakes as has been previously reported [30]. Inadequate dietary intake could have been overestimated by the dietary assessment over 7 days due to recall bias [49]. Our study confirms that the relationship between BMI and nutritional intake is not as strong as in the general population since Holvoet et al. report that about 50% of malnourished or at risk of malnutrition people in their hemodialysis population present with BMIs over 25 [50]. Fiber consumption in our study was lower than those reported in the literature [29,43,51].

There are several limitations to our study. The small size of the study population and particularly anuric patients restricted multiple analyses. The quantitative evaluation of the residual diuresis and/or the determination of toxins in the urine would have been useful. In addition, the food collection was carried out over 7 days. A measurement error in dietary recalls is unavoidable although efforts were made to minimize this with diet verification undertaken by a qualified renal dietician. The composition of the microbiota was not studied, so we cannot confirm the mechanism responsible for the decrease in the concentrations of UTs observed. Presumably, the use of laxatives could affect the production of IS and PCS as described in CKD rats where lactulose intake decreases the amount of IS and PCS [52].

To our knowledge, this study is the first to show the association between fiber intake and the protein/fiber index on IS and PCS concentrations in anuric HD patients. We confirm this result, which has been reported in non-dialysis CKD patients [21]. We demonstrate a negative correlation between the serum IS and PCS concentration and the fiber content in g/kg/day. Further interventional studies are warranted to determine the effectiveness of a decrease in the protein/fiber index by increasing fiber intake in achieving clinically important decreases in IS and PCS and improving patient outcomes. More than avoiding specific nutrients, a diet that decreases this protein/fiber index by increasing the intake of fibers could be a good option to decrease UT concentration and limit the risk of undernutrition.

## 4. Materials and Methods

### 4.1. Study Design

This analysis was conducted as part of Intra Individual Evaluation of Uremic Toxin Concentration (EVITUPH) an open-label monocentric prospective study that included 75 HD patients between 07/2015 and 09/2015 in Nephrology Service in Marseilles, France, with a one-year follow-up (registered at ClinicalTrials.gov NCT02480699). The initial study was performed to explore the intra-individual variability of uremic toxins over one year.

### 4.2. Study Population

Eligible participants were patients aged 18 or older, undergoing HD for more than 3 months. Inclusion criteria were that the subjects were capable of giving informed consent and agreeing to participate and were able to fill out a food questionnaire in French. Exclusion criteria were patients who had received an antibiotic during the month preceding their inclusion. All patients provided written consent. This clinical trial was funded by AP-HM, grant number AORC Junior 2014, and was approved by the French National Ethics Committee CPP (Comité de protection des Personnes Sud-Méditerranée I) on 31 March 2015 with 2015-A00319-40 reference in accordance with the precepts established by the Helsinki declaration. As the primary objective of EVITUPH was descriptive (i.e., description of the variability of UTs levels over 12 months), a calculation of the number of subjects needed was not necessary. We planned to include 75 hemodialysis patients. This number was estimated from published studies in the field of toxin level and dietary intake [26,28] that included a maximum of 56 patients. Seventy-five patients seemed relevant and feasible to study the impact of dietary intake on UT levels.

### 4.3. Dietary Intake Assessment and Biological Analysis

Blood was collected from all patients at the beginning of their middle-week HD session. Serum total concentrations of IS and PCS were analyzed by high-performance liquid chromatography (HPLC) using a fluorescence detection method [53].

Participants’ dietary intakes were assessed using a seven-day diet record that collected all their food intakes in quantity and quality over a period of seven days. The questionnaire was provided to all patients by a qualified renal dietician during one of their routine HD sessions within 15 days after their inclusion in the study. The dietician explained how to fill out the questionnaire and assess food quantities with visual proportion scales. If necessary, a family member was contacted. Seven days after having provided the patients with the questionnaire, the dietician verified whether the patient completed all parts of the questionnaire. If the patient had not completed or completed partially the questionnaire, a new one was offered, and renewed explanations were provided to the patient. After two unsuccessful attempts, the patient was considered non-contributory and was not included in the present study. This method is considered appropriate for capturing the usual intake over a one-month period. Dietary data were to be entered using the Nutrilog^®^ software, Marans, France (version 3.2). If necessary, the database was updated with patient recipes. Nutritional variables: energy (kcal), protein (g), fat (g), carbohydrates (g), protein of animal or vegetable origin (g), sugars (total, g), saturated fatty acids (%), and polyunsaturated fatty acids (%) were extracted from each questionnaire. The protein/fiber index was calculated by dividing the total protein intake (g) by the total fiber intake (g). 

### 4.4. Statistical Analysis

Patients’ characteristics were expressed as mean (standard deviation) for normally distributed continuous data, median (inter-quartile range (IQR)) for continuous data, and frequencies (percentages) for categorical data. All continuous variables were assessed for normality.

We used a *t*-test or chi-squared test to compare variables. Correlations between the uremic toxins and selected dietary intake variables were assessed using Spearman’s correlation coefficients. To highlight kidney clearance of uremic toxins, analyses were conducted in anuric patients. Univariable analysis was performed to find predictors of toxin concentration, then multivariable linear regression was completed to identify independent associations between dietary fiber, protein, the protein/fiber index, and toxins after adjustment for other known predictors of toxins including age, weight, dialysis vintage, and diuresis. Statistical analysis was performed using JMP^®^ 14.2.0, SAS Campus Drive, Cary, North Carolina 27513, USA. The null hypothesis was rejected at the 0.05 level.

## Figures and Tables

**Figure 1 toxins-14-00589-f001:**
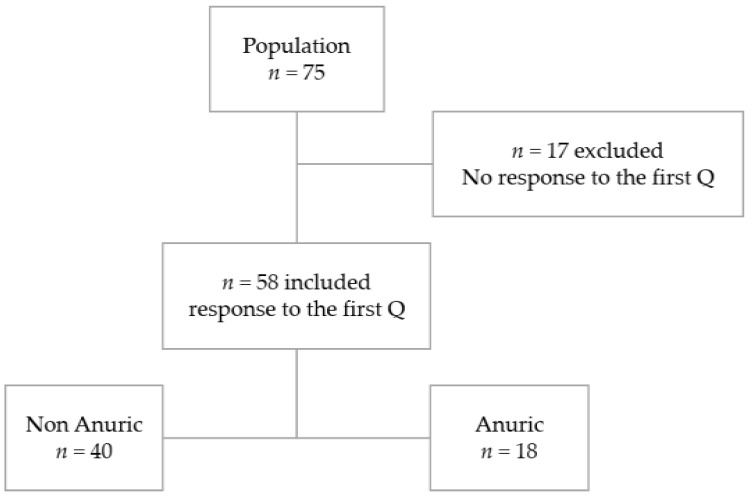
Flow diagram. Q: food questionnaire.

**Figure 2 toxins-14-00589-f002:**
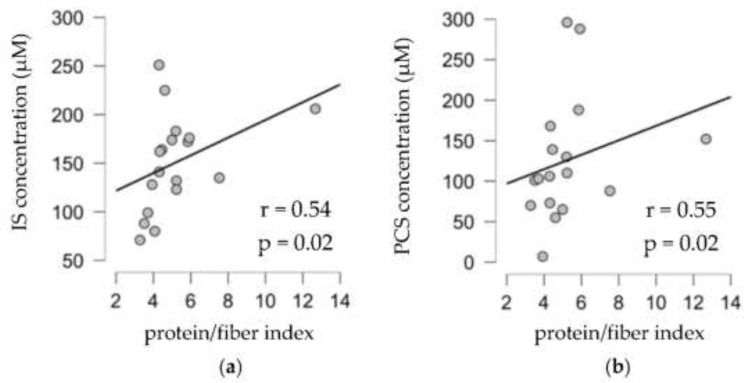
Correlation between indoxyl sulfate (IS) (**a**) and p-cresyl sulfate (PCS) (**b**) with protein/fiber index in anuric patients (*n* = 18).

**Table 1 toxins-14-00589-t001:** Clinical and biological characteristics of the patients who responded to the first nutritional questionnaire.

	All Patients *n* = 58 (%)	Non-Anuric *n* = 40 (%)	Anuric *n* = 18 (%)	*p*
Gender M/F (%)	37 (64)/21 (36)	25 (62.5)/15 (37)	12 (63)/6 (31)	1.000
Age (years)	66 ± 12	66 ± 13	64 ± 12	0.560
BMI (kg/m^2^)	25.5 ± 4.8	24.7 ± 4.1	27.3 ±5.6	0.048
Smoker current/past	23 (40)/7 (12)	14 (35)/5 (12)	9 (47)/2 (10)	0.549
Diabetes mellitus	19 (33)	15 (37)	4 (21)	0.366
Coronary heart disease	23 (40)	15 (37)	8 (42)	0.773
Peripheral vascular disease	18 (30)	14 (35)	4 (21)	0.377
Dialysis vintage (years)	5.1 ± 7.4	3.7 ± 7.4	8 ± 6.8	0.004
Vascular access AVF/PC	34 (59)/24 (41)	23 (58)/17 (42)	11 (61)/7 (39)	1.000
Serum creatinine (μmol/L)	787 ± 274	708 ± 233	956.8 ± 287	<0.001
Serum protein (g/L)	68.1 ± 4.8	68.1 ± 5.1	68.3 ± 4.3	0.907
Serum albumin (g/L)	37.4 ± 3.8	37.5 ± 4.2	37.4 ± 3.2	0.931
Bicarbonates (mmol/L)	19.5 ± 2.4	19.5 ± 2.3	19.5 ± 2.7	0.982
Phosphorus (mmol/L)	1.6 ± 0.5	1.6 ± 0.5	1.5 ± 0.4	0.421
Potassium (mmol/L)	4.9 ± 0.7	5.0 ± 0.8	4.8 ± 0.4	0.341
CRP (mmol/L)	14 ± 29	12 ± 20	19 ± 42	0.373
IS (µmol/L)	108.3 ± 54.1	91.5 ± 43	150.6 ± 49.4	<0.001
PCS (µmol/L)	130.0 ± 94.7	146.0 ± 97.3	125.8 ± 76.5	0.455

Data are presented as mean +/− standard deviation or number (%). *p*-values are calculated according to a *t* test between anuric patients
and non-anuric patients. M/F: male/female;
BMI: body mass index; AVF: arteriovenous fistula; PC: permanent catheter; CRP: C reactive protein; IS: indoxyl sulfate; PCS: p-cresyl sulfate. All biological data were assessed before the midweek dialysis session.

**Table 2 toxins-14-00589-t002:** Dietary intake of macronutrients in patients.

			All Patients *n* = 58	Non-Anuric *n* = 40	Anuric *n* = 18	*p*
Energy (kcal/kg/d)			21.9 ± 9.2	23.7 ± 9.5	18 ± 7.4	0.029
Protein (g/kg/d)			0.95 ± 0.3	1.0 ± 0.3	0.78 ± 0.3	0.005
Carbohydrates (g/kg/d)			2.3 ± 1.1	2.5 ± 1.1	1.9 ± 0.8	0.057
Fat (g/kg/d)			0.92 ± 0.4	1.0 ± 0.4	0.7 ± 0.4	0.025
PROTEIN	Animal protein	(g/d)	44.6 ± 13	46.9 ± 14.7	39.6 ± 6.9	0.048
		(g/kg/d)	0.64 ± 0.2	0.70 ± 0.2	0.53 ± 0.1	
	Plant protein	(g/d)	18.5 ± 7.4	19.0 ± 7.0	17.3 ± 8.1	0.004
		(g/kg/d)	0.28 ± 0.1	0.30 ± 0.1	0.24 ± 0.1	
Fiber (g/d)			14.6 ± 5.2	15.7 ± 5.4	12.6 ± 4.7	0.039
Fiber (g/kg/d)			0.2 ± 0.1	0.24 ± 0.12	0.17 ± 0.08	0.021
Protein/fiber index			4.9 ± 1.8	4.8 ± 1.7	5.2 ± 2.1	0.550
Animal protein/fiber index			3.5 ± 1.8	3.4 ± 1.7	3.7 ± 2.2	0.438
Plant protein/fiber index			1.3 ± 0.3	1.2 ± 0.3	1.4 ± 0.3	0.099
FAT	Saturated fat (g/d)		23.8 ± 9.5	25.2 ± 9.5	20.7 ± 8.8	0.097
	Unsaturated fat (g/d)		8.2 ± 4.7	8.3 ± 3.8	7.9 ± 6.4	0.734
Potassium (g/d)			2.2 ± 0.7	2.3 ± 0.7	1.9 ± 0.5	0.040

Data are presented as the mean ± SD.
*p*-values are calculated according to a *t* test between anuric and non-anuric patients (non-anuric).

**Table 3 toxins-14-00589-t003:** Correlation between indoxyl sulfate and p-cresyl sulfate and dietary components of energy, protein, and fiber intake in anuric patients (*n* = 18).

	Indoxyl Sulfate		p-Cresyl Sulfate	
	r	*p*	r	*p*
Age (years)	−0.23	0.226	0.32	0.208
Weight (kg)	0.45	0.056	0.03	0.859
Dialysis vintage	−0.36	0.135	−0.22	0.240
CRP	−0.03	0.900	−0.04	0.881
Energy (kcal/d)	−0.01	0.642	−0.58	0.014 *
Energy (kcal/kg/d)	−0.27	0.287	−0.43	0.168
Protein (g/d)	0.06	0.810	−0.37	0.134
Protein (g/kg/d)	−0.21	0.409	−0.22	0.586
Fiber (g/d)	−0.38	0.120	−0.63	0.013 *
Fiber (g/kg/d)	−0.56	0.015 *	−0.47	0.041 *
Protein/fiber index	0.54	0.021 *	0.55	0.037 *
Animal protein/fiber index	0.47	0.048 *	0.46	0.052
Plant protein/fiber index	0.06	0.823	−0.03	0.823

Data square roots are transformed prior to Spearman’s correlation for continuous variables and Student’s test for categorical variables. * *p* < 0.05.

**Table 4 toxins-14-00589-t004:** Multivariate analysis with linear regression between indoxyl sulfate and p-cresyl sulfate and age and protein/fiber index in anuric patients (*n* = 18).

	Indoxyl Sulfate		p-Cresyl Sulfate	
	Test	*p*	Test	*p*
Age (years)	−2.47	0.027 *	1.56	0.144
Dialysis vintage	−0.70	0.499	−1.09	0.296
Weight	1.99	0.066	−0.52	0.612
Protein/fiber index	2.57	0.022 *	0.01	0.991

Data square roots are transformed prior to multivariable linear regression. * *p* < 0.05.

## Data Availability

Not applicable.

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
