# Peer review of "Protein/Fiber Index Modulates Uremic Toxin Concentrations in Hemodialysis Patients"

_toxins, 2022, doi:10.3390/toxins14090589_

Round 1
Reviewer 1 Report
The paper describes protein/fiber index was positively correlated with Indoxyl sulfate (IS) and p-cresol sulfate (PCS) concentration in hemodialysis (HD) patients. This study shed light on association with fiber intake in patients with HD. The authors have some novel data but some of the methods need to be clarified
1. Is fiber intake beneficial for HD patients? Please clarify that (line 137-141).
2. Were intake of fiber or Protein/fiber index correlated with creatinine? Please clarify correlation between IS and/or PCS concentration and renal dysfunction.
Author Response
The paper describes protein/fiber index was positively correlated with Indoxyl sulfate (IS) and p-cresol sulfate (PCS) concentration in hemodialysis (HD) patients. This study shed light on association with fiber intake in patients with HD. The authors have some novel data but some of the methods need to be clarified
Inclusion and exclusion criteria are better described in methodology section
- Is fiber intake beneficial for HD patients? Please clarify that (line 137-141).
Answer: Adjonction of negatively or positively was added to clarify that the higher the fiber intake, the lower IS and PCS concentration
In the same way, the higher the protein to fiber ratio, the higher IS And PCS concentration.
- Were intake of fiber or Protein/fiber index correlated with creatinine? Please clarify correlation between IS and/or PCS concentration and renal dysfunction.
Answer: In our dialysis population (whole population or anuric population) creatinine was not correlated with Protein/fiber index. The correlation between UT and Protein fiber index is not explained by the renal dysfunction because in anuric patients, glomerular filtration rate is zero.
Reviewer 2 Report
Authors show an interesting study indicating a correlation between level of some uremic toxins and fiber intake in hemodialysis patients. The problem of the quality (low protein) diet in CKD remains controversial, especially in recent studies and due to lack of large trials. This manuscript focuses of IS and PCS level, two main UT, especially in uremic anuric patients, in whom UT accumulation and related complications can be more severe.
Minor remarks:
1) please explain abbreviations when used for the first time (i.e. UT line 23);
2) some minor language corrections are needed, i.e. 'nutritionnal' (line 10); 'data ere assessing' (line 76); 'vegetal' protein can be replaced by 'plant' protein (Table 2);
3) have you considered an analysis between UT, protein/fiber intake and inflammatory markers? and cardiovascular complications? some data are available in this subject (Xie LM, Ge YY, Huang X, Zhang YQ, Li JX. Effects of fermentable dietary fiber supplementation on oxidative and inflammatory status in hemodialysis patients. Int J Clin Exp Med. 2015 Jan 15;8(1):1363-9. eCollection 2015).
Author Response
Authors show an interesting study indicating a correlation between level of some uremic toxins and fiber intake in hemodialysis patients. The problem of the quality (low protein) diet in CKD remains controversial, especially in recent studies and due to lack of large trials. This manuscript focuses of IS and PCS level, two main UT, especially in uremic anuric patients, in whom UT accumulation and related complications can be more severe.
Minor remarks:
- please explain abbreviations when used for the first time (i.e. UT line 23);
Answer: Added in line 23
- Some minor language corrections are needed, i.e. 'nutritionnal' (line 10); 'data ere assessing' (line 76); 'vegetal' protein can be replaced by 'plant' protein (Table 2);
Answer: Done
- Have you considered an analysis between UT, protein/fiber intake and inflammatory markers? and cardiovascular complications? some data are available in this subject (Xie LM, Ge YY, Huang X, Zhang YQ, Li JX. Effects of fermentable dietary fiber supplementation on oxidative and inflammatory status in hemodialysis patients. Int J Clin Exp Med. 2015 Jan 15;8(1):1363-9. eCollection 2015).
Answer: In our study, we have C reactive protein as inflammatory markers. The association between IS and protein/fiber index is not modified with adjunction of CRP in the statistical model. (idem for multivariate analysis if CRP is implemented in the model.
Reviewer 3 Report
Dear author and editor:
This is an interesting paper discussing about how diet protein/fiber index may influence the serum uremic toxins. They propose that “IS and PCS concentrations were negatively correlated with fiber intake and positively correlated with protein/fiber index”. However, there are several limitations in this study.
First, as the authors mentioned in their limitation, the eligible participants are too few. They include 75 patients and nearly one-fourth of them dropped out, which might affect the final result greatly. Is there any anuric patient dropped out in the inclusion period (2 months, 2015/7 to 2017/9)? Uremic patients are prone to disease, did they drop out solely because of “cannot recall the diet record for 7 days” or did they suffer from disease, such as GI bleeding? How about the exclusion criteria? If there is any, please describe in the methodology.
They also reached the above mentioned conclusion by only 18 anuric patients who are under regular dialysis. As shown in the figure 2, the linear regression may reach a significant result due to an outlier. They should check the needed sample size before reaching the conclusion. Normally, it takes at least 30-40 patients to reach a statically significant result, depending on the power and alpha error.
Second, as the main topic discussing about the diet, the diet for the non-anuric and anuric patients are quite different in the first place. Table 2 showed anuric patients are having much lower energy, protein, fat, fiber, potassium but with a significant higher BMI (27 vs.24, p=0.048). How to interpret this result is quite against our common sense.
Besides, by table 1, the age of anuric patient are younger but having a significant longer dialysis vintage (8 years) compared with the non-anuric groups.
How the results we found in this small group (n=18) can apply to other anuric uremic patients under dialysis are quite questionable.
Minor issues are
1. The front page lack of the authors information and the source of IRB (which should be taken before study began)
2. The HPLC method for detection of indoxy sulfate and P-cresol sulfate should br mentioned in the supplement, as for other interested scientists to repeat their study.
3. The Table S1 may be considered to merge with table 4.
Author Response
Dear author and editor:
This is an interesting paper discussing about how diet protein/fiber index may influence the serum uremic toxins. They propose that “IS and PCS concentrations were negatively correlated with fiber intake and positively correlated with protein/fiber index”. However, there are several limitations in this study.
- First, as the authors mentioned in their limitation, the eligible participants are too few. They include 75 patients and nearly one-fourth of them dropped out, which might affect the final result greatly. Is there any anuric patient dropped out in the inclusion period (2 months, 2015/7 to 2017/9)? Uremic patients are prone to disease, did they drop out solely because of “cannot recall the diet record for 7 days” or did they suffer from disease, such as GI bleeding? How about the exclusion criteria? If there is any, please describe in the methodology.
Answer: In our population, 58/75 (77%) were included, “only” 17 did not completed the nutritional questionnaire. 2 partially completed and 15 did not complete the diet record stating that it was too demanding. The rate of complete questionnaires around 75% is usual for food questionnaires especially for 7 days questionnaire because that require a time investment for the patient. On the 17 patients who did not complete the questionnaire, 5/17(30%) were anuric.
Inclusion and exclusion criteria are better described in methodology section
- They also reached the above mentioned conclusion by only 18 anuric patients who are under regular dialysis. As shown in the figure 2, the linear regression may reach a significant result due to an outlier. They should check the needed sample size before reaching the conclusion. Normally, it takes at least 30-40 patients to reach a statically significant result, depending on the power and alpha error.
Answer: if we excluded the outlier data, the correlation between IS and protein fiber index persists with r =0.5 (p =0.04)
For PCS, correlations is no longer significant with r 0.46 and (p 0.06) but with a strong tendency
Second, as the main topic discussing about the diet, the diet for the non-anuric and anuric patients are quite different in the first place. Table 2 showed anuric patients are having much lower energy, protein, fat, fiber, potassium but with a significant higher BMI (27 vs.24, p=0.048). How to interpret this result is quite against our common sense.
Answer: One of the explanations is that as they are anuric, they have more restricted diet on potassium, water.
The relationship between BMI and nutritional intake is not as strong as in the general population. Holvoet et al reports that about 50% of malnourished or at risk of malnutrition in their hemodialysis population present with BMI over 25 (46).
Besides, by table 1, the age of anuric patient are younger but having a significant longer dialysis vintage (8 years) compared with the non-anuric groups.
Answer: In our population, age between the two groups is not different (p 0.55). However, it is true that the time spent on dialysis for these not-so-old patients is particularly long. This may be explained because this population was selected from a regional referral centre that cares for patients who are complicated with numerous comorbidities.
How the results we found in this small group (n=18) can apply to other anuric uremic patients under dialysis are quite questionable.
Answer : We confirm a previous study that protein-fibre index is associated to IS level in another CKD patient population. The choice to restrict our population to anuric dialysis patients is explained by the fact that in the absence of residual renal function, toxin levels can only be explained by production variations. In the dialysis population, serum creatinine is not associated with residual renal function but rather with protein nutritional status. We therefore decided to focus on anuric patients. Our paper reinforces the impact of diet on TU levels and represents a revolution in toxin reduction therapy. In recent years, studies aimed at reducing TU levels have focused on dialysis techniques and/or pro/pro/symbiotic strategy. With our study, as well as the results of Rossi et al, the role of dietary intake and the quality of dietary intake on UT level is reinforced. Reducing tryptophan and/or phenylalanine/tyrosine intakes, which seems the most obvious in view of UT production, is not necessarily the best solution. In fact, reducing tryptophan intake is tantamount to reducing protein intake, which is all the more complicated as patients often already have protein intakes below the recommended level (1.2g/kg/d). [30]. In addition, Wyant et al. recently identified another tryptophan-derived UT, kynurenic acid, which plays a protective role against ischemia. Reduced tryptophan intake could lead to a decrease in kynurenic acid and worsen ischemic disease, which is a major complication of CKD patients. Reducing deleterious UT such as IS or PCS while maintaining protective UT such as kynurenic acid by increasing fiber intake is certainly a better approach. Playing on the protein-fibre index seems much more interesting.
Minor issues are
- The front page lack of the authors information and the source of IRB (which should be taken before study began)
Answer: These elements were deliberately concealed during the submission process in order to respect the anonymity requested by the MDPI
- The HPLC method for detection of indoxyl sulfate and P-cresol sulfate should br mentioned in the supplement, as for other interested scientists to repeat their study
Answer: We used our HPLC method which we published in ref 51, so we have not put details in the supplement section. We can add them if necessary.
- The Table S1 may be considered to merge with table 4. In this study, authors investigated the association of the protein/fiber index and uremic toxins concentrations.
Answer: Table S1 refers to the total population, while Table 4 refers only to the 18 anuric patients.
Reviewer 4 Report
In this study, authors investigated the association of the protein/fiber index and uremic toxins concentrations.
Study is interesting, however, several issues should be addressed before potential publication:
Authors should emphasize better the gap in the literature this article is filling, as well as novelty of this study, with more according references.
In Methods, it should be emphasized where this study took place.
How was the final sample size decided on?
More information regarding study population could be added, including enrolment details.
P values should always be written in 3 decimal places.
Authors should expand the Discussion with more information regarding clinical implications that can be derived from the results of this study.
Author Response
Study is interesting, however, several issues should be addressed before potential publication:
Authors should emphasize better the gap in the literature this article is filling, as well as novelty of this study, with more according references.
In Methods, it should be emphasized where this study took place.
Answer: done
How was the final sample size decided on?
Answer: As the primary objective of EVITUPH was descriptive (i.e. description of the variability of UT levels over 12 months), a calculation of the number of subjects needed was not necessary. We planned to include 75 hemodialysis patients. This number was estimated from published studies in the field of toxin level variability that included a maximum of 56 patients. 75 patients seemed relevant and feasible to study the impact of dietary intake on TU levels
More information regarding study population could be added, including enrolment details.
Answer Inclusion and Exclusion criteria are better described in the methodology section.
P values should always be written in 3 decimal places
Answer: done
Authors should expand the Discussion with more information regarding clinical implications that can be derived from the results of this study.
Answer: done
Round 2
Reviewer 1 Report
The authors have made substantial revisions in response to the original critique.
Reviewer 3 Report
The number of this sudy is too small to reach a convincing result.
The authors did not show how they reach a effective sample size with only 18 patients.
Furthermore, the study population is quite heterogeneous (with anuric patients with muuch higher BMI and lower protein intake, while younger but with a longer dialysis vinatge)
If their focus is on anuric patients under hemodialysis, I will suggest they change the title and increase the sample size.
If their focus is on the whole dialysis population, their study population is too heterogenous to reach a conclusion.
(Besides, their exclusion criteria is only mentioned wih those who had antibiotic one month before the inclusion. For those who may have GI bleeding is not excluded by definition. Why they choose abtibiotic usage as exclusion criteria may need more explanation)
Sorry for rejecting this manuscript.
Reviewer 4 Report
No further comments.